# Clinical Diagnostic Imaging Study of Osteoradionecrosis of the Jaw: A Retrospective Study

**DOI:** 10.3390/jcm10204704

**Published:** 2021-10-14

**Authors:** Ikuya Miyamoto, Ryoichi Tanaka, Shintaro Kogi, Genki Yamaya, Tadashi Kawai, Yu Ohashi, Noriaki Takahashi, Mitsuru Izumisawa, Hiroyuki Yamada

**Affiliations:** 1Division of Oral and Maxillofacial Surgery, Department of Oral and Maxillofacial Reconstructive Surgery, Faculty of Dental Medicine, Iwate Medical University, Morioka 020-8505, Japan; kshinta@iwate-med.ac.jp (S.K.); gyamaya@iwate-med.ac.jp (G.Y.); kawait@iwate-med.ac.jp (T.K.); yohashi@iwate-med.ac.jp (Y.O.); yamadah@iwate-med.ac.jp (H.Y.); 2Division of Oral and Maxillofacial Radiology, Department of Oral and Maxillofacial Reconstructive Surgery, Faculty of Dental Medicine, Iwate Medical University, Morioka 020-8505, Japan; rtanaka@iwate-med.ac.jp (R.T.); tnoriaki@iwate-med.ac.jp (N.T.); mizumisa@iwate-med.ac.jp (M.I.)

**Keywords:** osteoradionecrosis, trismus, computed tomography, magnetic resonance imaging, positron emission tomography, single-photon emission computed tomography, diagnostic imaging

## Abstract

Radiation therapy (RT) plays a significant role in the management of head and neck malignancies. This study aimed to review the clinical symptoms and various imaging findings of osteoradionecrosis (ORN) and provide a clinical perspective on the development of ORN. The retrospective cohort was composed of 57 sites in 54 patients who had a history of RT and suspected ORN and 48 sites in 45 patients who were confirmed to have ORN. Image analyses included computed tomography (CT), magnetic resonance imaging (MRI), positron emission tomography (PET)/CT, bone scintigraphy, and single-photon emission CT (SPECT). The irradiated tissue was damaged by RT, and the extent of damage was correlated with clinical symptoms. The bone marrow showed sclerotic changes and the devitalized bone showed bone resorption after invasive stimulation. Chronic trismus and pathological fracture are considered severe conditions, typically occurring in the last stage of ORN. Furthermore, neurological symptoms were an important sign of tumor recurrence, since diagnostic imaging was difficult. The possible treatment options vary depending on the stage of ORN. We speculate that bone sclerosis reactions and bone resorption are sequential reactions that seem to be protective measures of the bone to radiation injury.

## 1. Introduction

Radiation therapy (RT) plays a significant role in the management of head and neck malignancies. Patients with such malignancies usually experience the full spectrum of collateral damage from RT (such as xerostomia, chronic trismus, dysgeusia, dysphagia, and decreased tongue mobility) [1]. Osteoradionecrosis (ORN) is a problematic complication that occurs when irradiated bones become devitalized [2]. ORN is a challenge for clinicians.

The mechanism of pathogenesis is unclear; however, the most frequently reported theory is radiation arteritis, which is the concept of hypoxia, hypovascularity, and hypocellularity [2]. In recent years, Delanian et al. proposed the theory of radioactive fibrosis [3,4,5,6,7]. Bone necrosis is a common characteristic of ORN and has clinical signs and symptoms [8,9]. Typical osseous findings of ORN on computed tomography (CT) include cortical disruption, disorganization of trabeculation, and osseous fragmentation. Furthermore, in the irradiated field, it can be associated with significant soft tissue thickening and enhancement in the adjacent masticator muscles [10,11]. Diagnostic imaging for osteomyelitis has changed with the development of diagnostic imaging equipment [12]. Imaging findings of ORN are common and specific, but few comprehensive evaluations have been performed clinically with several modalities [13,14,15].

This retrospective study aimed to comprehensively review the clinical symptoms and various imaging findings of ORN and to provide a clinical perspective on the development of ORN in the jawbone.

## 2. Materials and Methods

This retrospective consecutive case series study was conducted at the Division of Oral and Maxillofacial Surgery, Iwate Medical University, Japan, following the tenets of the Declaration of Helsinki (1964). The study design was approved by the Ethics Committee of the Faculty of Dental Medicine, Iwate Medical University (01296). All clinical data were obtained through a review of medical charts and data from other related hospitals.

From 57 sites in 54 patients who had a history of RT and suspected ORN, 48 sites in 45 patients who were confirmed to have ORN according to the definition of ORN after various clinical and imaging examinations were performed between 1 January 2010 and 30 June 2021. The excluded patients were two tumor recurrences and eight dental disorders. RT was performed not only at our university hospital but also at all other institutions.

ORN is usually defined as exposed irradiated bone tissue that fails to heal over a period of three months without residual or recurrent tumors [15,16]. However, this definition did not include the infrequent condition of bone appearance on radiological examination in the presence of intact oral mucosa or facial skin [17,18]. Bone necrosis originating from the bone marrow enables complete mucosal coverage [19]. Therefore, the definition of ORN by Støre and Boysen, as “radiological evidence of bone necrosis within the radiation field, where tumor recurrence has been excluded”, is simple and easy to understand [20]. Based on the clinical features of ORN, patients were classified according to the He classification of ORN [17].

The duration of follow-up covered the period from the completion of RT to the day of the first visit to our department. As patients developed ORN during the research period, the specific sites affected were recorded. The clinical manifestations were summarized, including bone exposure, skin fistula, limitation in mouth opening, and pathological fractures. The maximum diameter of the bone lesion and the status of the pathological fracture were assessed by multi-detector CT (MDCT) and cone-beam CT (CBCT); the maximum diameter was determined by the maximum length acquired from the CT images. Bony changes on conventional images were evaluated with reference to the CT appearance. However, it was impossible to compare the Hounsfield unit (HU) between pre- and post-treatment on CT images due to the difference between MDCT and CBCT. At our institution, these modalities are standard examinations for patients undergoing follow-up for malignancy. All imaging examinations including CT, MRI, ^18^F-fluorodeoxyglucose positron emission tomography/CT (^18^F-FDG-PET/CT), bone scintigraphy and single-photon emission CT (SPECT) were reviewed retrospectively in a random order, separated by two certified oral and maxillofacial surgeon and radiologist (I.M. and M.I.), blinded to the clinical findings, the original study interpretation. A total of 48 CT, 42 MRI, 35 PET/CT, 4 bone scintigraphy, and 14 SPECT images were evaluated.

### 2.1. MDCT and CBCT

MDCT examinations of the maxillofacial bones were performed using 64-detector multi-detector row CT scanners (GE Healthcare, Boston, MA, USA). CT images were obtained from the top of the lower orbit through the mandible. CT studies were performed by helical scanning using a pitch of 0.96, 120 kVp, and 24.0-cm Field of view (FOV). Axial image reconstruction was performed with bone and standard algorithms, and 2.5-mm thick sagittal and coronal multiplanar reformatted images were obtained. CBCT was performed using a 3D Accuitomo F17 (J. Morita Corp., Kyoto, Japan). CBCT images were obtained using a 0.28-mm voxel size.

The reviewers were asked to comment on the presence or absence of the following imaging findings:Sclerosis of the affected boneResorption of the affected boneA periosteum reaction of the affected bone

They were instructed that surrounding inflammatory changes (i.e., amorphous soft tissue stranding or infiltration) and muscle or tissue thickening did not qualify as a solid mass and a discrete, measurable mass must be present.

### 2.2. MRI

All MR images were acquired using a 3.0-Tesla system (MR750; GE Healthcare, Boston, MA, USA) with a circular polarized neck coil to visualize the level of the maxilla and mandible. MR images were obtained using the following four sequences: (1) short inversion-time inversion recovery (STIR); (2) fast spin echo T1 weighted images; and (3) fast spin echo T1 weighted images with contrast-medium enhancement.

The section thickness was 5.0 mm. The acquisition matrix was 320 × 224 pixels. Axial and coronal images were obtained. The classification of images and comparison with pathological findings were performed according to a previous study by Ariji et al. [21]. Briefly, T1 WI were classified into two patterns—low and no change. The image changes in the bone marrow were assessed by comparing the signal intensities (SI) of the contralateral side. T2 WI were classified into three patterns—homogeneous high, heterogeneous high, and homogeneous low type. A high pattern was defined as a strong and widespread increase in the SI. A low pattern was defined as a decline in the SI.

### 2.3. FDG-PET/CT

All patients fasted for at least 4  h prior to intravenous administration of 0.1  mCi (3.7  MBq) kg^−1^ body weight of ^18^F-FDG. Following the administration, the patient rested quietly for a standard 60-min uptake period, after which imaging was performed. Each patient underwent a single integrated PET-CT examination. Patients were instructed not to chew or talk during the examination. Prior to scanning, all patients removed objects such as dentures. The patients were positioned in the head-first, supine position. They were instructed to perform breath-holding during CT acquisition, which was performed first from the head to the pelvic floor. The study was performed using 16-section PET/CT scanners (Discovery 600 Motion, GE Healthcare, Boston, MA, USA). The CT scan parameters were 120 kVp, variable/smart milliampere, and 3.75-mm collimation. CT scanning was performed from the top of the skull through the abdomen. Following CT, PET data were acquired using a 4-min bed position. The PET/CT scanner had a bismuth germanium oxide scintillation crystal. The PET acquisition included an online delayed coincidence subtraction to correct for random coincidences as well as dead-time correction. All PET acquisitions were uniform using two-dimensional (2D) techniques. The radiologists were asked to record the maximum standardized uptake value (SUV_max_) around the ORN in some cases. The SUV_max_ was defined as the highest pixel value in the region of interest [22]. These parameters were obtained from a 2D region of interest placed on the axial image based on visual inspection.

### 2.4. Bone Scintigraphy and SPECT

All patients fasted for at least 4  h prior to intravenous administration of 555–740 MBq of ^99m^Tc-MDP in four bone scintigraphy and 14 SPECT cases respectively. Imaging settings for CT acquisition were 140 kV with a matrix size of 128 × 128 (slice thickness 1.9 mm) for the multi-slice CT (Infina Hawkeye 4, GE Healthcare, Boston, MA, USA).

### 2.5. Treatment

Treatment records were collected cautiously, including information on conservative therapy (irrigation and antibiotic prescription), hyperbaric oxygen (HBO) therapy, sequestrectomy, and segmental mandibulectomy with or without vascularized tissue reconstruction. The treatment outcomes were recorded as ‘resolved’, ‘improved’, ‘stable’, or ‘progressed’, as suggested by He et al. [17]. ‘Resolved’ relates to the status in which the patient is asymptomatic and has relatively normal function. ‘Improved’ relates to the status in which the patient has relief from the symptoms and necrotic lesions decrease on radiography. ‘Stable’ indicated that the disease has neither progressed nor improved. ‘Progressed’ designates a deterioration in the condition of the patient.

## 3. Results

### 3.1. Clinical Findings

Patient characteristics are summarized in Table 1. There were 36 male patients and 9 female patients (mean age, 68.6 years; age range, 39–85 years). Lesions were present at 39 sites in the mandibles and at nine sites in the maxilla, and three patients had both mandibular and maxilla lesions.

The initially affected tumor sites were the pharynx (19 (42.2%) patients), tongue (11 (24.4%) patients), gingiva (four (8.9%) patients), parotid gland (two (4.4%) patients), hard palate (two (4.4%) patients), buccal mucosa (two (4.4%) patients), floor of the mouth (two (4.4%) patients), nasal cavity (one (2.2%) patient), and maxillary sinus (one (2.2%) patient).

The tumor types included squamous cell carcinoma (SCC) (37 (82.2%) patients), adenoid cystic carcinoma (ACC) (one (2.2%) patient), malignant lymphoma (one (2.2%) patient), lymphoepithelial tumor (one (2.2%) patient), undifferentiated carcinoma (one (2.2%) patient), and plasmacytoma (one (2.2%) patient). Three (6.7%) patients did not have enough data. They were irradiated at a mean of 64.3 Gy, ranging from 40 Gy to 95 Gy. Thirty-seven (82.2%) patients were administered combined chemotherapy, six (13.3%) patients did not undergo chemotherapy, and two (4.4%) patients did not have enough data.

During chemotherapy, 13 (28.9%) patients underwent super-selective arterial chemotherapy. The tumor stage was stage I (two (4.4%) patients), stage II (one (2.2%) patient), stage III (four (8.9%) patients), stage IV (26 (57.8%) patients), and other (three (6.7%) patients), and no data (nine (20%) patients). The initial ORN classification by He et al. was stage 0 (four (8.9%) patients), stage I (15 (31.3%) patients), stage II (25 (52.1%) patients), and stage III (four (8.9%) patients). However, there were cases in which symptoms progressed during the observation period; therefore, the classification was revised to stage 0 (four (8.9%) patients), stage I (13 (27.1%) patients), stage II (25 (52.1%) patients), and stage III (six (12.5%) patients). The clinical features and outcomes of patients in the distinct stages of ORN are shown in Table 2. The initial clinical events of ORN were local bone exposure (19 (39.6%) patients), periodontal infection (14 (29.2%) patients), inadequate healing after tooth extraction (nine (18.8%) patients), dental implant placement (one (2.1%) patient), bone surgery (one (2.1%) patient), and unknown cause (four (8.3%) patients). The mean occurrence of ORN was 54.9 months (range, 0–168 months) after irradiation (Table 3).

Trismus was observed in 19 (39.6%) patients. Abnormal T2 signal intensity, enhancement, and thickening of the masseter and pterygoid muscles were associated with ORN (Table 4). The more advanced the ORN stage, the more trismus appeared. After reconstructive surgery, the symptoms of trismus improved.

### 3.2. CT Findings

On CT images, homogeneous bone consolidation was observed in 10 (20.8%) patients. Heterogeneous bone consolidation was observed in 30 (62.5%) patients, and six (12.5%) patients showed bone resorption. Using pre-treatment CT data, we compared the levels of bone consolidation. The cancellous bone was sclerosed in 28 (58.3%) sites; however, eight (16.7%) sites did not show clear bone consolidation images, and one (2.1%) site clearly showed bone resorption. Eleven (22.9%) sites did not have sufficient data (Figure 1a,b). Moreover, the periosteal reaction was not observed in 44 (91.7%) sites, and the periosteal reaction was observed in four (8.3%) sites. At first, there were four pathological fracture cases during the observation period, and finally, there were six cases.

### 3.3. MRI Findings

All 42 (100%) patients showed T1 WI hypo-intensity in the symptom region (Figure 2a,c,e). In contrast, the T2 WI showed various aspects. Thirty (71.4%) patients with T2 WIs showed homogeneous hyper-intensity (Figure 2b), which showed a relatively mild inflammatory condition of bone marrow clinically. Heterogeneous hyperintensity was observed in 11 (26.2%) patients (Figure 2d). One patient (2.4%) showed homogeneous hypointensity (Figure 2f).

### 3.4. PET/CT Findings

PET/CT showed uptake around the symptomatic bone in 30 (85.7%) patients. The images of five (14.3%) patients did not show abnormalities, and 13 (37.1%) patients did not have data. PET/CT revealed a mean SUV_max_ of 7.69 (*n* = 14) in the inflammatory ORN area. No uptake was observed in the sequestrum bone area.

### 3.5. Bone Scintigraphy and SPECT Findings

Seventeen (94.4%) of the measured patients showed uptake of ORN lesions, and one (5.6%) patient did not show uptake, which means that there was no uptake in the sequestration bone area.

The overall various radiographic characteristics of ORN are shown in Table 5.

### 3.6. Treatment Outcome

Conservative treatment was composed of saline irrigation and antibiotic prescriptions for 35 (72.9%) sites. Tooth extraction was performed at two (4.2%) sites. Infected bone or sequestrum removal was performed in five (10.4%) patients. HBO therapy was administered to seven (14.6%) patients 10–30 times. Mandibular saucerization was performed at one (2%) site. Hemimandibulectomy without reconstruction was performed at two (4.2%) sites. Mandibular segmental resection with vascularized fibular grafts was performed in three (6.3%) patients. The overall treatment outcome was resolved in eight (16.7%) cases, improved in 22 (44.8%) cases, stable in 15 (31.3%) cases, and progressed in three (6.2%) cases (Table 6).

## 4. Discussion

This study aimed to evaluate the diagnostic aspect of ORN with several imaging modalities and clinical findings to help clinicians understand the symptoms of ORN. ORN, a man-made disease developing after RT, was first observed in the early 20th century. ORN was first described by Regaud in 1922, and Ewing was the first to use the term ‘radiation osteitis’ to describe changes in the bone after RT in 1926 [18,23,24]. Ewing reported that after severe external irradiation, the bone was nearly devoid of circulation, and the periosteal vessels were generally ‘sclerosed’.

In this study, the cohort had an almost advanced stage of malignancy; therefore, chemotherapy was combined with RT in 82% patients. For oral cancer, RT is often performed in recurrent or uncontrollable cases [25,26]. In our institution, 13 patients underwent super-selective intra-arterial infusion therapy with RT owing to advanced disease [27,28]. There were 42.2% cases of pharyngeal cancer. RT is a standard treatment for oropharyngeal cancer, and the irradiation field includes the jawbone, especially the angle lesion [29]. The incidence of ORN may increase when chemotherapy is added to RT [30,31].

The occurrence of ORN is not time dependent; hence, it may become evident even years after RT [32]. ORN typically develops with a small incomplete wound healing and a small area of mucosal collapse with exposure of the underlying bone and reactive inflammatory granulation tissue, by triggers such as tooth extraction. This means ORN begins with traumatic invasion into the injured devitalized bone with mucosal collapse. As ORN progresses, patients often develop chronic inflammatory trismus, neuropathic pain, chronic drainage, and pathological fracture. In this study, approximately 50% patients complained of initial symptoms with dentistry-related events, and they could have been a trigger for ORN. Therefore, in clinical situations, dentists must inquire about the medical history of the patient carefully, particularly in oropharyngeal cancer, which is unfamiliar to general dentists, and mandibular molar extraction or dent-alveolar surgery should be performed carefully.

In ORN staging, there were no cases of external skin fistulas in stages 0 and I. In stage II, almost all cases were of intraoral mucosal defects. Some cases had intraoral mucosal defects and extraoral skin fistulas. These cases were pre-conditioned to stage III, which indicates a pathological fracture. In stage III, there were usually cases of thorough-and-through defects. These conditions may be related to the extent of the necrotic bone region. Extensively damaged bone, such as cortical disruption or fragmentation, has a way to the surrounding soft tissue and the bone inflammation spreads to the masticatory muscles by this route, thus indicating trismus.

From a diagnostic CT imaging point of view, 82% of the bone marrow showed sclerotic changes on CT. These results are similar to those reported by Alhilali et al. [22]. A possible explanation for the bone sclerosing mechanisms is that the damage from radiation to the bone tissue continues to stimulate bone cells. The damage affects osteocytes and activates osteoblasts, which cause reactive bone consolidation, particularly in the cancellous bone area. RT reduces not only the proliferation of bone marrow and periosteal and endothelial cells but also the production of the extracellular matrix, particularly the collagen [32]. In histological studies, the irradiated bone was usually devitalized with few osteocytes. Takahashi et al. investigated the morphological changes in the Harversian system after RT in rabbit femurs. Four weeks after radiation, there was occlusion of the Harversian vessels and dilation of capillaries with resorption of the perivascular bone matrix by osteoclasts [33]. Moreover, the absence of osteocytes may induce mineralization within the pericellular space and further spread into the lacuna, which ends in a consecutive deposition of minerals and, thus, hyper-mineralized lacunae [34]. These phenomena are similar to those of asymptomatic chronic mandibular osteomyelitis in the aged population, showing chronic stimulation of the impacted teeth to the cortical and cancellous bone in the intraoral space through the periodontal ligament and osteoconsolidation images with reduced osteocytes radiologically and histopathologically [35]. The condition seems to be similar to ‘micropetrosis’ [36]. In chronic osteomyelitis, micropetrosis is frequently observed, and in ORN, consolidated bone with micropetrosis is also observed. Therefore, sclerosed bones on RT indicate devitalized or reduced vitalized bones. In addition, the severely injured bone might be sequestrum, which results in bone exposure or resorption with inflammation. In ORN, the energy from radiation would be stronger than that from stimulation of ordinary chronic osteomyelitis; therefore, the clinical symptoms of ORN would be more severe than those of chronic osteomyelitis [10]. The sclerosis reaction is considered the response of bone tissue to radiation stimulation. In terms of bone response to RT, it is speculated that these reactions seem to be bone protective reactions against radiation injuries.

Moreover, 90% CT images showed few periosteal reactions in the irradiated area. These results suggest that RT affects not only the targeted bone tissue but also the surrounding soft tissues, particularly the periosteum, terminating the normal periosteum reactions. Similar results were reported by Ogura et al., which could be distinguished from medication-related osteonecrosis of the jaw (MRONJ). MRONJ was often observed periosteum reaction and the periosteum condition of MRONJ is usually less damaged than that of ORN [37,38].

MRI showed that the damage to the bone marrow by RT continued even after the long-term asymptomatic phase. Even if there are no clinical symptoms of ORN, the bone marrow is considered abnormal for a long time after RT. Considering the CT images, the consolidated bone region would have devitalized or reduced vitality, possibly fibrosis with lower blood supply. Partial bone resorption and reactive fibrosis showed a mixed image of bone consolidation (low intensities on T1 WI and heterogeneous hyper-intensities on T2 WI), as Kaneda has previously suggested [13,21]. Moreover, the homogeneous hyper-intensities of T2 WI may be mild bone marrow inflammation considering the clinical symptoms of ORN. From a clinical point of view, tooth extraction within the radiation field should be performed with caution at any time after RT. If abnormal bone marrow is invaded by triggers, the symptoms of ORN would change from the chronic inflammation phase to the acute inflammation phase. Tooth extraction before RT is recommended, but it does not prevent ORN; the essence of ORN is devitalized bone. Considering soft tissue, CT and MRI often show inflammation in the surrounding masticatory muscles, and MRI can confirm the inflammatory symptoms around the masticatory muscles [39]. These symptoms are related to trismus. As the ORN stage progressed, chronic trismus increased, which indicated that inflammation from the bone marrow extended beyond the cortical bone to the inferior border of the mandible and into the surrounding soft tissue. Trismus was observed with cortical disruption on CT (Figure 3a,c) and abnormal T2 signal intensity, enhancement, and thickening of the masseter and pterygoid muscles (Figure 3b,d). In such a condition, antibiotics only cause acute inflammation in the chronic phase. The degree of trismus would improve with surgical resection of the necrotic bone, however, with only conservative treatments, it is unclear whether trismus can be improved.

PET/CT findings of ORN reflect the disease course and indicate the point of clinical remission in patients with chronic osteomyelitis [40,41,42]. PET is effective in detecting tumor recurrence. Bone scintigraphy has been used to diagnose ORN; however, there is no quantitative index to evaluate ORN activity. In this study, we obtained information on the existence of ORN; however, the detailed situation of ORN could not be evaluated because of the low level of specificity [13]. Since the development of bone SPECT can aid in various quantitative SUV analyses, SPECT/CT is a promising modality for the analysis of ORN [42].

The same lesions captured by PET/CT and SPECT are shown in Figure 4a,b, respectively. PET/CT is useful for differentiating tumor recurrence. On the other hand, SPECT provides useful information regarding bone activity. PET/CT appeared to represent glucose metabolism, including the bone marrow, which was hyper-metabolized, and SPECT appeared to represent areas of increased bone metabolism; PET reflected inflammatory soft tissue and/or tumor recurrence and SPECT showed inflammatory and/or neoplastic bone tissue. PET/CT and SPECT are sensitive indicators of altered osteoblastic activity, but local disturbances in vascular perfusion, clearance rate, permeability, and chemical binding also affect imaging [13]. It is occasionally difficult to differentiate soft tissue uptake from bone uptake in patients with known cellulitis and possible underlying osteomyelitis. In terms of tumor recurrence, PET/CT and SPECT is useful but not definitive. Incisional biopsy is required for the final diagnosis.

ORN can occur spontaneously due to periodontal disease and apical lesions, possibly after injury induced by dental prosthesis and tooth extraction [43,44]. This could be explained by the continuity between the teeth and the devitalized bone through the periodontal ligament and periosteum. The imaging data suggest that the damage to the irradiated bone cannot heal to the normal bone marrow; it will continue as a chronic inflammatory condition. Invasive stimulation triggers ORN initiation.

Therefore, even small ulcers caused by dental prosthesis will not epithelialize the underlying devitalized bone due to its reduced biological activity, which may lead to the development of ORN with acute or chronic osteomyelitis. In addition, such bone conditions result in incomplete wound healing due to poor blood flow, and reactive fibrosis occurs in the bone marrow with bone resorption. Furthermore, the risk of ORN after invasive surgery outside the field of radiation is almost non-existent [43,44].

Therefore, the most important diagnostic point of ORN is the accurate region of the abnormal bone. An appropriate preoperative imaging evaluation of a lesion’s localization and extent is a key in treatment. Clinically, to grasp abnormal bone conditions with several imaging modalities, the combination of CT and MRI will provide the most effective information for ORN in this study.

In case of the progression of ORN to the inferior alveolar nerve, sensory neurological symptoms or numb chin syndrome appear [45,46]. This symptom also occurs in odontogenic origin, tumor recurrence, metastatic tumors, and MRONJ. Therefore, patients with the symptoms require careful examination that combines PET, SPECT, and biopsy in addition to CT and MRI. Among the cases excluded in the present study, there was a case in which diagnostic imaging was difficult to judge tumor recurrence or ORN, and neurological symptoms were an important sign of tumor recurrence.

In recent years, methods, such as intensity-modulated RT or 3D-conformal radiotherapy, have been used to reduce adverse events of RT; the results are favorable, and the procedures are promising [47].

Considering these backgrounds, a possible explanation of gradual bone destruction in ORN is as follows:Osteoconsolidation of cancellous bone due to irradiation with hypovascularity with reduced bone cell viabilities and bone marrow changes to fibrous or scar tissue. Bone and bone marrow are chronic inflammatory conditions.Progression of sclerosed cancellous bone inflammation and/or bone resorption.Bone invasion, such as tooth extraction, induces acute inflammation (acute osteomyelitis and ORN) and continuous bone resorption.Through cortical disruption or extensive bone destruction with refractory inflammation, the hard, fragile jawbone due to bone consolidation develops pathological fracture, intra-/extra-oral fistula, and soft tissue scars with trismus, which is the terminal stage of ORN, with the progression of local inflammatory bone resorption.

The schematic sequential mechanism of bone reaction after RT is shown in Figure 5.

The conservative treatment of ORN is fundamental, but its cure is limited. If symptoms continue, the basic strategy of ORN should be the surgical removal of the devitalized bone. HBO therapy was considered a common treatment, but randomized controlled trials concluded that HBO therapy was ineffective, particularly in advanced stages [48,49,50,51]. We have also administered HBO therapy in some cases, but its effects are unclear. This therapy alone seems ineffective in ORN because it cannot revitalize the dead bone [32]. The combination of pentoxifylline and vitamin E (PENTO) has become a new conservative treatment method gradually adopted by more clinicians [3,4,5,6,7]. The meta-analysis of PENTO suggested that it is effective for the treatment of ORN; however, no single therapy can help achieve a 100% cure of ORN, and each therapy has different therapeutic effects on ORN in different periods [7]. Therefore, various treatment methods can be selectively combined to achieve satisfactory therapeutic effects in patients. Concerning the site of abnormal signals for stages 0 and I, conservative treatments such as saline irrigation, administration of antibacterial agents, and bone scraping at the time of sequestration are effective. HBO therapy is also considered effective at these stages [48]. There seems to be a controversy regarding stage II. The surgical removal of abnormal bone is expected to improve clinical symptoms. However, this surgical stimulation may exacerbate the clinical symptoms in cases of extensive bone necrosis, and transition from stage II to stage III can occur. It is difficult to estimate the extent of bone resorption progression. The results of this study suggest that bone sclerosing images with abnormal signals on MRI are indicative of a high risk of surgical bone resorption. Conservative treatment for stage III disease is unlikely to improve the clinical condition. Invasive surgery or reconstruction surgery should be planned. It can be said that vascularized bone graft is the definitive treatment. When reconstructive surgery is difficult for various reasons, invasive jaw resection is effective. Although bone reconstruction has been proposed as a treatment choice, patients often refuse due to surgical invasion. Moreover, from a reconstruction point of view, it is technically difficult to reconstruct mandibular condyle head lesions. If conservative treatment is effective, it should be promoted; however, no matter how many ineffective treatment methods are continued, the burden on the patient will only increase. With sufficient physical condition and patient consent, surgical reconstruction is the fundamental solution. Our study has several limitations. First, the data for our study were collected from a single university in Japan and the people in biased area may have some cultural and geographical features. Second, we did not qualify the data because patients were included from not our department but also from other institution. Finally, the relatively small sample size limited the capacity for further stratified analysis. Therefore, well-designed, prospective, multicenter studies are required to validate the pathogenesis of ORN and effective treatments.

## 5. Conclusions

In this study, the bone, bone marrow and soft tissue in ORN were damaged with RT, and the extent of damage correlated with clinical symptoms and several clinical symptoms of ORN were associated with imaging modalities. The possible treatment options vary depending on the stage of ORN. It is important to make a comprehensive diagnosis based on the clinical condition and the findings of imaging types and modalities. The bone sclerosis reaction and subsequent bone resorption are considered the response of bone tissue to stimulation of RT. In terms of a jawbone response to RT, it is speculated that these sequences of reactions seem to be bone protective measures against radiation injuries.

## Figures and Tables

**Figure 1 jcm-10-04704-f001:**
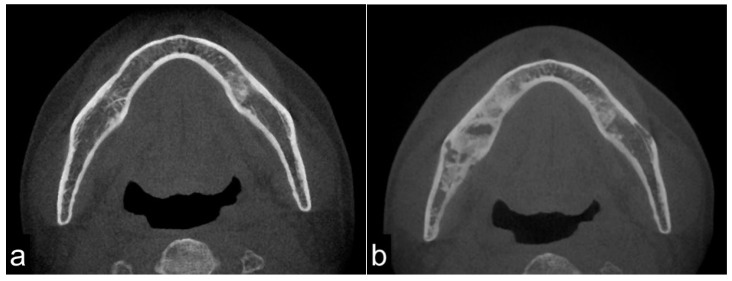
Pre-treatment mandible on axial computed tomography (CT) scan (**a**) and osteoradionecrosis (ORN) mandible on axial CT scan (**b**). The image shows bony sclerotic changes with ORN (**b**) involving the right mandible compared to the pre-treatment mandible in the same lesion without periosteum reaction (arrowhead). These reactions may have been induced by irradiation.

**Figure 2 jcm-10-04704-f002:**
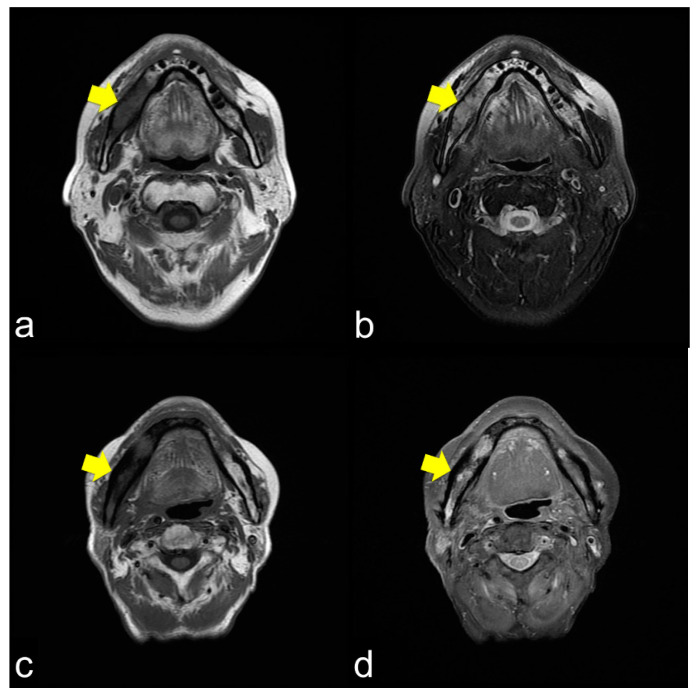
(**a**) Axial T1-weight image (T1 WI) shows low signal intensity from the molar lesion to the ramus on right side of the mandible (arrowhead); (**b**) Axial T2-weight image (T2 WI) reveals high signal intensity on the right side of the mandible (arrowhead). The margins between the normal and abnormal bone marrow are not distinct. This condition usually shows mild inflammation of the ORN; (**c**) Axial T1 WI shows low signal intensity from the molar lesion to the ramus on the right side of the mandible (arrowhead); (**d**) Axial T2 WI reveals heterogeneous intensity on the right side of the mandible (arrowhead). The margins between the normal and abnormal bone marrow are not distinct. This condition usually results in severe ORN inflammation; (**e**) Axial T1 WI shows low signal intensity from the molar lesion to the ramus on the right side of the mandible (arrowhead); (**f**) Axial T2 WI reveals low signal intensity on the right side of the mandible (arrowhead). The margins between the normal and abnormal bone marrow are not distinct. This condition is bone necrosis, which shows less inflammation.

**Figure 3 jcm-10-04704-f003:**
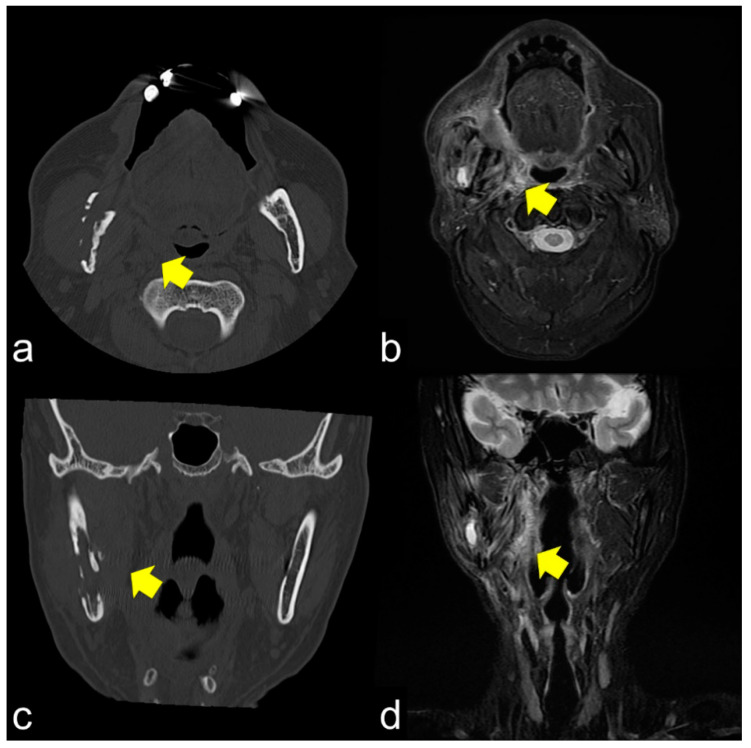
(**a**,**b**) Axial CT and T2 WI magnetic resonance images show cortical bone disruption and abnormal signal intensity in the masticatory muscle; (**c**,**d**) Coronal images also show chronic inflammation, particularly in the medial pterygoid muscle. Chronic inflammation in the masticatory muscles results in trismus.

**Figure 4 jcm-10-04704-f004:**
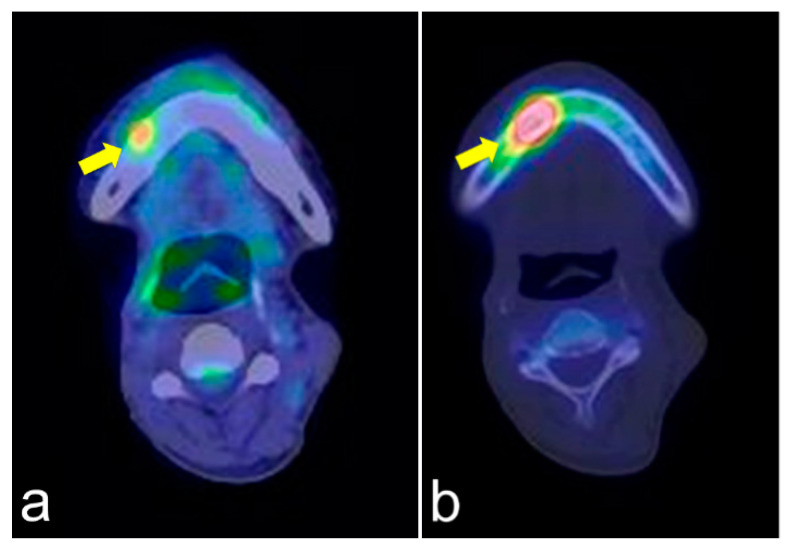
PET/CT (**a**) and single-photon emission computed tomography (SPECT) (**b**) images of the same lesion. PET/CT clearly shows soft tissue inflammation around the ORN area, while SPECT specifically shows bone inflammation. This comparison would help detect tumor recurrence.

**Figure 5 jcm-10-04704-f005:**
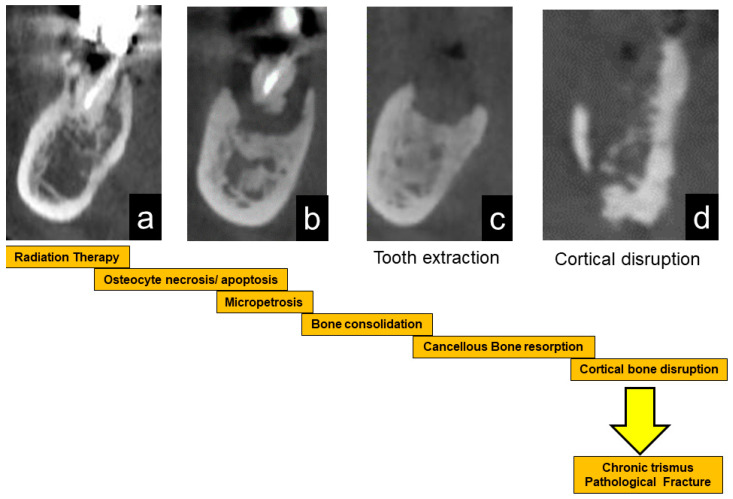
Schematic illustration of the progression of ORN. After radiation therapy, bone cells are damaged, leading to bone sclerosis (**a**). After that, the necrotic bone part is exposed (**b**). Stimulation such as tooth extraction causes infection and resorption of necrotic bone (**c**). When the cortical bone ruptures, inflammation spreads to the surrounding soft tissue, resulting in trismus (**d**). Eventually, it causes a pathological fracture.

**Table 1 jcm-10-04704-t001:** Patient and tumor characteristics.

	Characteristics	Numbers of Patients (%)
Gender	Male	36 (80%)
	Female	9 (20%)
Age Mean (range)	68.6 (39–85) years	
Maxilla/mandible		9 (18.8%)/39 (81.2%)
Primary tumor	Oropharyngeal	19 (42.2%)
	Tongue	11 (24.4%)
	Gingival	4 (8.9%)
	Parotid gland	2 (4.4%)
	Hard palate	2 (4.4%)
	Buccal mucosa	2 (4.4%)
	floor of the mouth	2 (4.4%)
	Nasal cavity	1 (2.2%)
	Maxillary sinus	1 (2.2%)
Histology	Squamous cell carcinoma	37 (82.2%)
	Adenoid cystic carcinoma	1 (2.2%)
	Undifferentiated carcinoma	1 (2.2%)
	Malignant lymphoma	1 (2.2%)
	Lymphoepithelial tumor	1 (2.2%)
	Plasmacytoma	1 (2.2%)
	No data	3 (6.7%)
Mean radiation dose (range)	64.3 (40–95) Gy	
Chemotherapy	(+)	37 (82.2%)
		including 13 (28.9%) patients treated with super-selective arterial chemotherapy
	(−)	6 (13.3%)
	No data	2 (4.4%)
Tumor stage	Stage I	2 (4.4%)
	Stage II	1 (2.2%)
	Stage III	4 (8.9%)
	Stage IV	26 (57.8%)
	Other	3 (6.7%)
	No data	9 (20.0%)
Initial events of osteoradionecrosis	Local bone exposure	19 (39.6%)
	Periodontal infection	14 (29.2%)
	Tooth extraction	9 (18.8%)
	Dental implant placement	1 (2.1%)
	Bone surgery	1 (2.1%)
	Unknown cause	4 (8.3%)

**Table 2 jcm-10-04704-t002:** Clinical classification of ORN by He et al.

ORN Stage	Number of Patients (%)
Stage 0	*n* = 4 (8.3%)
B0 S0: No evident signs or only osteolytic images on radiography; however, the patient suffers from typical ORN-related symptoms (bone exposure or pain).	100%
Stage I	*n* = 15 (31.3%) → 13 (27.1%)
B1 S0: The maximum diameter of the lesion on radiography is <2 cm, and there is no mucosa or skin defect.	*n* = 3 (6.3%)
B1 S1: The maximum diameter of the lesion on radiography is <2 cm, and there is an intraoral mucosa defect or external skin fistula alone.	*n* = 12(25.0%) → 10 (20.8%)
B1 S2: The maximum diameter of the lesion on radiography is <2 cm, and there is a through-and-through defect.	*n* = 0 (0%)
Stage II	*n* = 25 (52.1%)
B2 S0: The maximum diameter of the lesion on radiography is >2 cm, and there is no mucosa or skin defect.	*n* = 2 (4.2%)
B2 S1: The maximum diameter of the lesion on radiography is >2 cm, and there is an intraoral mucosa defect or external skin fistula alone.	*n* = 20 (41.7%)
B2 S2: The maximum diameter of the lesion on radiography is >2 cm, and there is a through-and-through defect.	*n* = 3 (6.3%)
Stage III	*n* = 4 (8.3%) → 6 (12.5%)
B3 S0: A pathological fracture is identified on radiography, and there is no mucosa or skin defect.	*n* = 1 (2.1%)
B3 S1: A pathological fracture is identified on radiography, and there is an intraoral mucosa defect or external skin fistula alone.	*n* = 0 (0%) → 2 (4.2%)
B3 S2: A pathological fracture is identified on radiography, and there is a through-and-through defect.	*n* = 3 (6.3%)

**Table 3 jcm-10-04704-t003:** The mean onset duration for ORN after RT.

ORN Stage	Mean Duration ± Standard Deviation (Months)
Stage 0 (*n* = 4)	90 ± 25.8
Stage I (*n* = 15)	40.1 ± 44.0
Stage II (*n* = 25)	64.8 ± 42.9
Stage III (*n* = 4)	14 ± 16.7
Overall (*n* = 48)	54.9 ± 44.1

**Table 4 jcm-10-04704-t004:** Percentage of trismus in the ORN stage.

ORN Stage	Number of Patients (%)
Stage 0	1/4 (25%)
Stage I	2/13 (15%)
Stage II	11/25 (44%)
Stage III	5/6 (83%)
Overall	19/48 (39.6%)

**Table 5 jcm-10-04704-t005:** Radiological findings of CT, MRI, PET/CT, and bone scintigraphy.

Modality		Findings	Number of Patients (%)
CT	Bone mineral condition	No bone resorption	*n* = 2 (4.2%)
		Cancellous homogeneous Bone sclerosis	*n* = 10 (20.8%)
		Cancellous heterogeneous Bone sclerosis	*n* = 30 (62.5%)
		Bone resorption	*n* = 6 (12.5%)
		Periosteum reaction	(+) *n* = 4 (8.3%)
			(−) *n* = 44 (91.7%)
MRI	Bone marrow viability	T1 hypo-intensity	*n* = 42 (100%)
		T2 homogeneous intensity	*n* = 30 (71.4%)
		T2 heterogeneous intensity	*n* = 11 (26.2%)
		T2 hypo-intensity	*n* = 1 (2.4%)
PET/CT	Glucose metabolic activity	Active	(+) *n* = 30 (85.7%)
			mean SUV_max_ = 7.69 (*n* = 14)
		Non-active	(−) *n* = 5 (14.3%)
Bone scintigraphy/SPECT	Bone metabolic activity	Active	(+) *n* = 17 (94.4%)
		Non-active	(−) *n* = 1 (5.6%)

CT, computed tomography; MRI, magnetic resonance imaging; PET, positron emission tomography; SPECT, single-photon emission CT.

**Table 6 jcm-10-04704-t006:** Overall treatment outcome.

Treatment Outcome	Number of Patients (%)
Resolved	*n* = 8 (16.7%)
Improved	*n* = 22 (45.8%)
Stable	*n* = 15 (31.3%)
Progressed	*n* = 3 (6.2%)

## Data Availability

Not applicable.

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
