# Peer review of "Clinical Diagnostic Imaging Study of Osteoradionecrosis of the Jaw: A Retrospective Study"

_jcm, 2021, doi:10.3390/jcm10204704_

Round 1
Reviewer 1 Report
In this manuscript, Miyamoto and coauthors reviewed the clinical symptoms and various imaging findings of osteoradionecrosis and provided a clinical perspective on the development of ORN in the jawbone. However, there are still some important limitations in this study, such as the subject sample number is small and the data for this study were collected from a center. Although these absence, the overall manuscript are well written and organized
Author Response
Dear Sir,
there are still some important limitations in this study, such as the subject sample number is small and the data for this study were collected from a center.
--Thank you very much for your suggestion. As you pointed out the number of participants is limited and further prospective research is needed. Therefore, we added the sentences in Page 15 465-466.
Therefore, well-designed, prospective, multicenter studies are required in order to validate the pathogenesis of ORN and effective treatments.
Thank you very much.
Ikuya Miyamoto.
Reviewer 2 Report
The manuscript submitted to JCM entitled “Clinical diagnostic imaging study of osteoradionecrosis of the jaw: A retrospective study” is an original article which aim to review the clinical symptoms and various imaging findings of osteoradionecrosis (ORN) and to provide a clinical perspective on the development of ORN in the jawbone.
On my opinion the article is interesting, well written, with good English.
However, I highlighted some issues.
- English language: Minor spell check is required.
- Abstract: Please structure the abstract to attract the reader's attention and adapt it accordingly.
- Introduction: This section has been properly prepared.
- Material and methods: This section has been properly prepared.
- Results: This section has been properly prepared.
- Discussion: Please discuss the onset of numb chin syndrome (or mental neuropathy) as predictive factor of ORN onset also considering radiological features [https://doi.org/1016/j.jormas.2018.04.006].
- Conclusions: This section has been properly prepared.
After making the indicated changes, the article will be suitable for publication.
Thanks for the opportunity to review this manuscript.
Author Response
Dear Sir,
Thank you very much for your quick reviewing of our manuscript.
1) Abstract: Please structure the abstract to attract the reader's attention and adapt it accordingly.
--Thank you very much for your advice. We changed some part of the abstract with more concreate results to attract the reader’s attention.
“The bone marrow showed sclerotic changes and the devitalized bone showed bone resorption after invasive stimulation. Chronic trismus and pathological fracture are considered a severe condition, typically in the last stage of ORN. Furthermore, neurological symptoms were an important sign of tumour recurrence, since diagnostic imaging was difficult.”
2) Discussion: Please discuss the onset of numb chin syndrome (or mental neuropathy) as predictive factor of ORN onset also considering radiological features [https://doi.org/1016/j.jormas.2018.04.006].
--Thank you very much for your suggestion. Neurological symptoms were a clinically significant sign. We added the literatures and the experience of our excluded patient who was difficult to judged tumour recurrence or ORN.
Thank you very much for your valuable suggestion.
Ikuya Miyamoto